# Exploring ways to respond to rising obesity and diabetes in the Caribbean using a system dynamics model

Leonor Guariguata[1]*, Leandro Garcia[2], Natasha Sobers[1], Trevor S. Ferguson[3], James Woodcock[4], T. Alafia Samuels[5], Cornelia Guell[6], Nigel Unwin[1,4,6]*

1 George Alleyne Chronic Disease Research Centre, University of the West Indies, Cave Hill, Barbados, 2 Centre for Public Health, School of Medicine, Dentistry and Biomedical Sciences, Queen's University, Belfast, United Kingdom, 3 Epidemiology Research Unit, Caribbean Institute for Health Research, University of the West Indies, Mona, Jamaica, 4 MRC Epidemiology Unit, University of Cambridge School of Clinical Medicine, Cambridge, United Kingdom, 5 Caribbean Institute for Health Research, University of the West Indies, Mona, Jamaica, 6 European Centre for Environment and Human Health, University of Exeter Medical School, Cornwall, United Kingdom

* leonor.guariguata@gmail.com (LG); nigel.unwin@mrc-epid.cam.ac.uk (NU)

**Data Availability Statement:** The data and model are fully available at: An online version of the model is available via Silico: https://silico.app/@

## Abstract

Diabetes and obesity present a high and increasing burden of disease in the Caribbean that have failed to respond to prevention policies and interventions. These conditions are the result of a complex system of drivers and determinants that can make it difficult to predict the impact of interventions. In partnership with stakeholders, we developed a system dynamics simulation model to map the system driving diabetes and obesity prevalence in the Caribbean using Jamaica as a test case. The study aims to use the model to assess the magnitude changes necessary in physical activity and dietary intake to achieve global targets set by the WHO Global Action plan and to test scenarios for interventions to reduce the burden of diabetes and obesity. Continuing current trends in diet, physical activity, and demographics, the model predicts diabetes in Jamaican adults (20+ years) to rise from 12% in 2018 to 15.4% in 2030 and 20.9% by 2050. For obesity, it predicts prevalence to rise from 28.6% in 2018 to 32.1% by 2030 and 39.2% by 2050. The magnitude change necessary to achieve the global targets set by the World Health Organization is so great as to be unachievable. However, a combination of measures both upstream (including reducing the consumption of sugar sweetened beverages and ultra processed foods, increasing fruit and vegetable intake, and increasing moderate-to-vigorous activity) at the population level, and downstream (targeting people at high risk and with diabetes) can significantly reduce the future burden of diabetes and obesity in the region. No single intervention reduces the prevalence of these conditions as much as a combination of interventions. Thus, the findings of this model strongly support adopting a sustained and coordinated approach across various sectors to synergistically maximise the benefits of interventions.

lguariguata/diabetes-model?s= i8NOUzgpTHuxQIXmyqc-Qw The model was developed using R with "deSolve" package. The Git repository is available at: https://github.com/ leoguari/SSCH-Model.

**Funding:** This work was directly supported by a joint grant from the Wellcome Trust and UK Medical Research Council (MR/N005384/1). NU was the grant recipient. In addition, the work benefitted from the activities of two projects based in the Caribbean, led by the University of the West Indies, and funded by the Canadian International Development Research Centre. The funders had no role in study design, data collection and analysis, decision to publish, or preparation of the manuscript.

**Competing interests:** The authors have no competing interests to declare.

## Introduction

Rising prevalence of obesity and type 2 diabetes have been of concern for decades [1, 2]. There have been a series of calls to action, including the WHO Global Action Plan for the Prevention and Control of NCDs 2013–2020 [3] with global targets set for no increase in obesity or diabetes by 2025 compared to 2010. The relationship between obesity and diabetes is well-established [4] and many of the trials designed to prevent or delay the onset of type 2 diabetes have set weight loss as the cornerstone of interventions, usually through a combination of calorie restriction and increased physical activity in those at highest risk of progressing to type 2 diabetes [5]. While these have been shown to be highly effective in controlled settings, they have not halted the inexorable increase in diabetes prevalence at the population level. The most likely cause for this failure is not because of the intervention design, but because the system driving obesity and subsequent diabetes is more complex than is represented in a controlled study or trial [6]. One way to characterise complex systems is through mapping of the feedbacks, delays, flows and accumulations (stocks) [7]. The complex nature of systems can make it hard to predict their behaviour and the effects on outcomes. In 2006, Jones et al. developed a system dynamics model for exploring the likely progression of the diabetes and obesity epidemics in the United States and tested scenarios for intervening in that system to reduce the burden of diabetes and its complications [8]. The model was developed using a complex systems approach engaging directly with key stakeholders to map and simulate the dynamics of the epidemic and led to the informed development and adoption of policies to target the epidemic [9].

The Caribbean is faced with a particularly high double burden of type 2 diabetes and obesity [10], driven in large part by an increasingly sedentary population [11] and unhealthy diets [12]. It is expected that this high burden will increase cases and deaths over the coming decades. As early as 2007, Caribbean policymakers recognized the importance of targeted and coordinated responses [13], but despite this the prevalence of obesity and diabetes has continued to rise [14]. As a result, the System Science in Caribbean Health project was developed with the overarching aim of using systems thinking and modelling to help inform, through engaging with stakeholders, coordinated interventions designed to curb the rise in obesity and type 2 diabetes [15]. The study also aims to enable more realistic targets and measures of success to be developed. We present a system dynamics model that is part of that project to simulate the burden of diabetes and its relationship with obesity, including inputs for diet and physical activity to help understand the possible trajectories of the epidemics, simulate the impact of changing determinants, and understand the magnitude of change needed to achieve at least a stabilisation in diabetes and obesity in adults in one exemplar Caribbean island, Jamaica.

## Methods

### Ethics statement

The study was submitted for ethical review and approved by the University of the West Indies, Cave Hill Institutional Review board, and also approved by the Barbados Ministry of Health Research Ethics Committee.

In addition, ethical approval for the stakeholder consultations was obtained from the University of the West Indies, Mona, Jamaica, Ethics Committee; the University of Medicine and Health Sciences, St. Kitts and Nevis; and the Ministry of Health, Wellness, and the Environment, St. Vincent and the Grenadines with a waiver of informed consent from participants.

## Summary of the study

The model presented here is one component of a larger study [15] with multiple mixed methods components. The study used methods from system dynamics to engage stakeholders in the development of conceptual maps, or causal loop diagrams (CLDs) [16], to describe the drivers of rising obesity and diabetes prevalence in the Caribbean. Key stakeholders were identified and gathered from experts in the region representing a breadth of experience in healthcare, academia, civil society, government, intergovernmental organisations, food producers, retailers and distributors in the Caribbean. Experts in food science and physical activity were also included in the sessions and through consultations to inform the development of CLDs. Stakeholders participated in in-depth interviews and group model building (GMB) sessions where they were tasked with developing the conceptual maps, identifying feedback cycles, and elaborating potential systems-oriented interventions to change the trajectory of diabetes and obesity in the region. The details and outputs for these workshops have been previously described [17, 18]. Those diagrams and discussions form the basis for the development of the simulation model presented here.

## Study setting

Jamaica was selected as a test case for the simulation model. This was in part because it has a history of repeated population-based national cross-sectional health surveillance surveys [19–22] that provided data inputs as well as measures for calibration. It is the largest English speaking island in the Caribbean with an evidence base for noncommunicable diseases (NCDs) trends [1] and an engaged political profile for NCD prevention and management [23] that often serves as a model for other countries in the region [24]. Jamaica was considered by stakeholders to provide a case study relevant to other English speaking countries in the Caribbean.

## Model structure

System dynamics models are nonlinear time-continuous simulation models that use first-order differential equations and integrals that are connected using a causal structure(16). The causal structure for the model was developed and informed by the outputs from the GMB sessions with key stakeholders in the region. The CLDs developed by stakeholders helped to inform the summary stock and flow diagram presented here (see Fig 1), and the full model diagrams presented in the S1 File. Some of the model structure adapts aspects of the model developed by Jones et al. [8] but, in following key stakeholder input, expands the connections for upstream determinants including diet and physical activity. It also includes differences in distribution of risk factors and diabetes by gender, an important dimension of NCDs in the Caribbean [25]. We chose a time scale from 1990 and projected forward to 2050 in one year time steps.

 The model structure follows a core stock (as boxes) and flow (thick double-lined arrows with valves) structure that represents the movement of people through different glycemic states: normoglycemia (NG), prediabetes (preDM), and diabetes (DM). Added up, these stocks account for the whole of the adult population (20+ years) with no overlap between stocks. There is an inflow of adults for population growth into the normoglycemic stock. People may move from normoglycemia to prediabetes and back to normoglycemia. We assume those with prediabetes may also move into diabetes, and no population-level remission from diabetes. For each stock there is also an outflow for all-cause mortality. The flows are modified by other factors including the ageing of the population, the change in obesity prevalence, and for onset of diabetes; and sub-models for dietary intake (including the consumption of sugar sweetened beverages (SSBs), fruit and vegetable intake), and physical activity (PA).

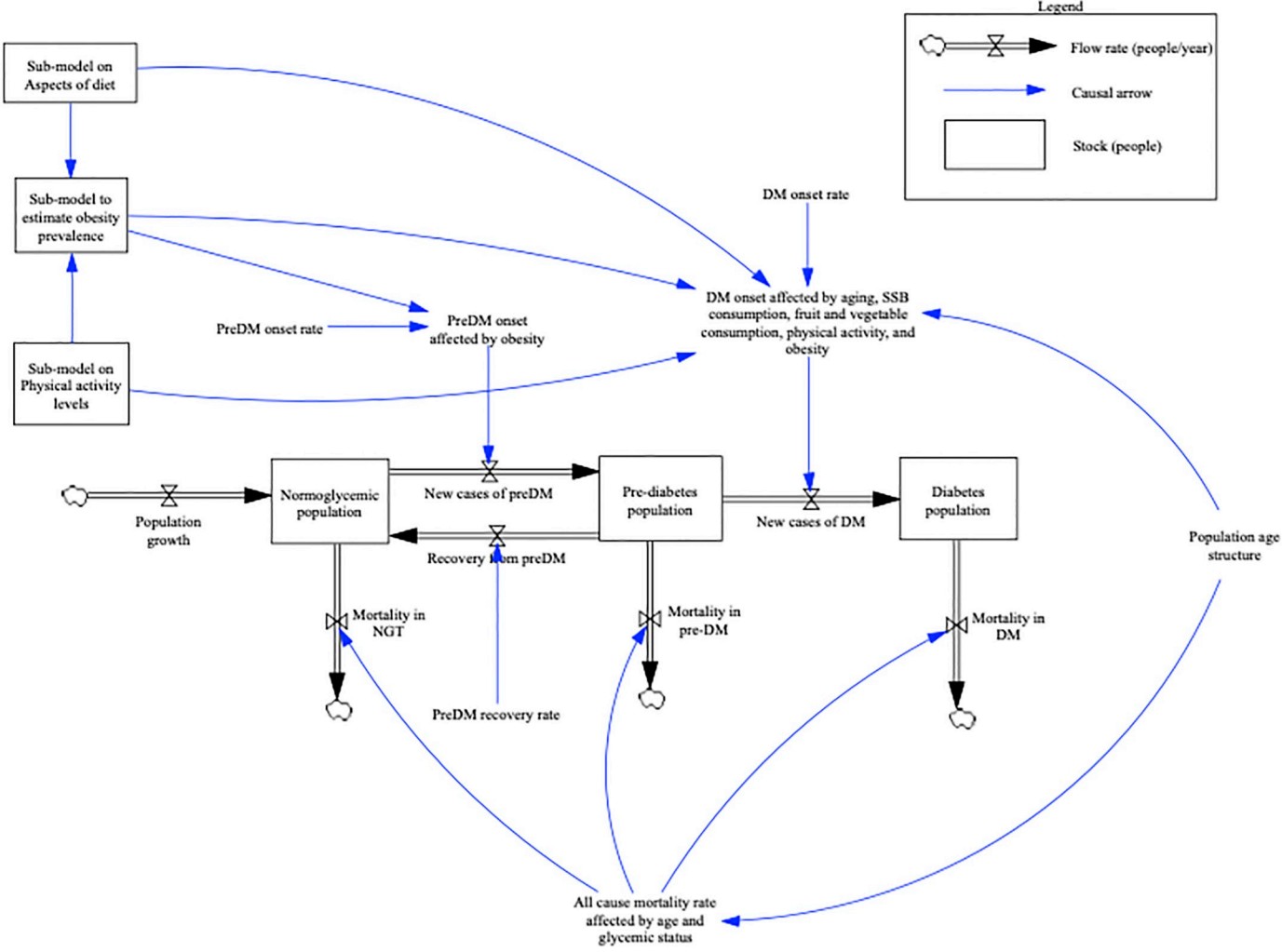

**Fig 1. Summary stock and flow diagram of the core structure of the model for diabetes in the Caribbean.**

## Model parametrisation and calibration

The approaches taken here were guided by system dynamics methodology described by Sterman [16]. This model is a reflection of the priorities of stakeholders engaged in the GMB as well as an iterative and collaborative process of testing and revision grounded in the scientific literature. It is, however, a deliberate simplification and seeks to model priorities identified by key stakeholders and for which there was sufficient scientific evidence. We parameterized and calibrated the model using historical trends of diabetes and obesity, and selecting Jamaica as a test case.

A full list of data inputs including time series, calculations, and a detailed accounting of data sources is available in the S1 File. Key data sources and inputs are listed in Table 1.

As far as possible, data sources were chosen that were most closely related to the population of interest. We prioritised data inputs and information for testing assumptions from studies conducted in Jamaica, then in the Caribbean region, and finally in the African diaspora or pooled estimates from other countries or systematic reviews with meta-analysis.

**Table 1. Key data inputs\*.**

| Data input | Source | Summary |
|---|---|---|
| Population inflows, age structures | UN World Population Prospects—2019 Revision [26] | Net population inflow to the normoglycemic stock |
| Baseline estimates of prediabetes and diabetes prevalence and population mean body mass index (BMI) | Jamaica Health and Lifestyle Surveys [19–21], Spanish Town study [22], ICSHIB Cohort [27] | Cross-sectional surveys conducted in Jamaica at various time points |
| Flow (incidence) rates for prediabetes and diabetes onset, and prediabetes remission | Scientific literature [28–30] | Estimates were taken from systematic reviews of longitudinal cohort studies studying the progression of hyperglycemia in various populations |
| Physical activity estimates | Interpolation using modelled trends from Ng and Popkin [31] and Jamaican study data [20, 32] | Physical activity estimates were taken from the literature for studies done in Jamaica and trends interpolated |
| Dietary intake including sugar sweetened beverage consumption | Scientific literature and studies conducted in the Caribbean [33–38] | Interpolated data from studies conducted in Jamaica, Barbados and Latin America |
| Effect sizes for interventions and risk factors | Scientific literature (see S1 File) | Relative risk estimates and effect sizes for interventions taken from systematic reviews and meta-analyses in the literature |

\* see S1 File for a full discussion of the inputs, data sources, and time series

Where data were not available, we used mean values and pooled estimates from systematic reviews and calibrated inputs based on plausibility and alignment with the historical data available for measured diabetes prevalence and obesity prevalence from Jamaica [25]. We estimated body mass index (BMI) and obesity prevalence from interpolated trends in physical activity and from historical trends in diet. Physical activity estimates were derived from a combination of data points for physical activity obtained from device-based and self-reported cross-sectional surveys in Jamaica [16, 28] and trends and projections estimated from Ng and Popkin [31] to project likely scenarios for decreasing physical activity. Dietary intake estimates were similarly interpolated from data in Jamaica [36] and Barbados [33–35] and combined with global dietary trends [37, 39] and studies done in Latin American countries on ultra-processed food intake [38]. We assume that trends in the future for dietary intake patterns follow a similar trajectory to the past. A detailed description of adjustments and interpolations is available in the S1 File. These were used to estimate caloric intake and expenditure. We use the Harris-Benedict equations [40] to estimate resting metabolic rates and the Forbes constant [41] to estimate weight changes over time from mean caloric imbalance per person in a year in the adult population to arrive at changes in body mass for men and women. The BMI estimates for men and women are used to estimate the prevalence of obesity which has a direct impact on the flow rates for prediabetes and diabetes onset.

We then used the effects of changes over time in obesity prevalence, and population age structure to estimate the incidence of prediabetes in the population and added effects for the change in sugar sweetened beverage intake, fruit and vegetable intake [42], and physical activity for estimating the incidence of diabetes [43] (see Fig 1). Changes in total caloric intake are not modelled to have an independent effect on diabetes incidence. Any reduction in caloric intake will only change diabetes incidence through the change in weight. For dietary intake, SSB intake and fruit and vegetable intake were modelled with an effect on diabetes incidence independent of any effect on caloric balancer (Fig 1, S1 File). We also included an independent effect on total physical activity on diabetes incidence.

## Model implementation

The model was developed using the online Silico App [44], and in R [45] using the deSolve package [46] and methods described by Duggan [47]. Diagramming was done using Vensim

PLE Software [48]. The S1 File provides information on how to access the full model and inputs online. Many of the equations for the calculations of prediabetes and diabetes onset and mortality were adapted from Jones et al. [8] and for obesity by models by Homer et al. [49]. A full accounting of the equations used in the model, sensitivity analyses, and a discussion of how these were developed is available in the S1 File.

### Developing scenarios for testing

The objective of the model is to create a test environment for simulating interventions and changes in determinants necessary to substantially shift diabetes prevalence and obesity prevalence in Jamaica by 2050. To do this we developed three sets of scenarios: testing the necessary changes to achieve global targets for NCDs; scenarios guided by stakeholder determined priorities; and evaluating the impact of single interventions.

### Achieving global targets

The first scenarios tested were the magnitude of change necessary to achieve the targets set by the World Health Organization Global Action Plan (GAP) for the Prevention and Control of NCDs [3] in 2013, which sets as an objective to stop the increase in prevalence of diabetes and obesity among adults by 2025 using 2010 as the baseline. We modified physical activity (total moderate-to-vigorous physical activity (MVPA)) and caloric intake to understand the effect on obesity and diabetes prevalence (%) in adults. We assumed that physical activity continues to decrease over time in line with trends and projections by Ng and Popkin [31] and that caloric intake increases over time in line with global dietary trends [39]. We modelled changes in duration of MVPA and relative percentage decreases in caloric intake from 2010 to 2025. We applied a magnitude change so that the relative trend in decreasing physical activity and increasing caloric intake are maintained. For example, if MVPA is doubled, we simply double the whole of the trend line but assume a continued decreasing trend of MVPA into the future.

### Scenarios relevant to the Caribbean

In consultation with stakeholders both from group-model building sessions in the region and with local experts contributing to this study, we tested a number of scenarios that are of interest to the region and were considered plausible public health strategies for preventing and managing diabetes and obesity. A description of the interventions and combinations of interventions is presented in Table 2 and in the S1 File. Scenarios for obesity reduction do not include any downstream interventions like bariatric surgery because there is no evidence that these have a measurable impact at the population level. As with the scenarios above, we applied a magnitude change in determinants to the inputs so that the relative trends are maintained. Alternative trends and interventions were tested in complementary analyses that are presented in detail in the S1 File.

### Individual policy scenarios

We modelled the effects of the individual scenarios included in the combined scenarios presented in Table 2 on diabetes prevalence and obesity. For these, we constructed dose-response curves which can be found in the S1 File.

### Sensitivity analyses

We conducted sensitivity analyses for diabetes prevalence modifying input variables for assumptions related to incidence, mortality, and effect sizes for interventions. These were

**Table 2. Combined and individual model scenarios for decreasing diabetes and obesity prevalence in adults.**

| Scenario | Details | Effect sizes and assumptions |
|---|---|---|
| **Combined interventions** | | |
| Downstream | Targeted interventions of people at high risk of or living with diabetes.<br>• Assumes 25% of the pre-diabetes population receives and adheres to a lifestyle intervention to reduce diabetes incidence.<br>• One-third of people with diabetes receive and adhere to diabetes self-management education and leads to a reduction in all-cause mortality. | • Pooled 19% reduction in diabetes incidence from lifestyle modification interventions [5]<br>• Pooled 25% reduction in all-cause mortality in people with diabetes receiving self-management education [50] |
| Modest upstream | A combination of diet and physical activity measures.<br>• A 15% reduction in SSBs consumption<br>• A 15% reduction in consumption of other ultra processed foods<br>• A 15% increase in fruit and vegetable consumption<br>• Front-of-package warning labels (FOPL)<br>• An additional 15 minutes of MVPA per day<br>• Public information campaigns on healthy diet and physical activity | • A 26% increase in diabetes incidence per unit per day consumption of SSBs [42]<br>• A 4% reduction in diabetes incidence per serving per day consumption of fruits and vegetables [42]<br>• FOPLs lead to a 6.6% mean reduction in caloric intake, 13% increase in fruit and vegetable intake and a 13% decrease in ultra processed foods [51]<br>• Up to 4g/day increase in consumption of fruits and vegetables from an economic model [52]<br>• Pooled relative risk increased MVPA in those targeted by 28% [53] |
| Intensive upstream | The same interventions as the modest upstream, but with greater intensity.<br>• A 25% reduction in SSBs consumption<br>• A 25% reduction in consumption other ultra processed foods<br>• A 25% increase in fruit and vegetable consumption<br>• Front-of-package warning labels (FOPL)<br>• An additional 30 minutes of MVPA per day<br>• Public information campaigns on physical activity and healthy diet | The effect sizes are the same as those assumed from the moderate intensity interventions. |
| Combined downstream and intensive upstream | A combination of the downstream interventions and the intensive upstream interventions described above | A combination of the downstream and the intensive upstream effects described above |

done using Latin Hypercube Sampling for multi-variable and univariable analyses. A detailed description and the results of these analyses are available in the S1 File.

## Results

### Baseline predictions

Fig 2a–2c presents the output simulated by the model in the historical period of 1990 to 2020 and projecting into the future to 2050 for diabetes prevalence (%), obesity prevalence (%), and mean BMI (kg/m$^2$) (for men and women) in adults, and includes historical survey produced estimates obtained from Jamaican studies(19–22). The predictions assume trends in determinants will continue on their current trajectories so that the population is expected to age (as described by the World Population Prospects) [26], caloric intake, in particular the intake of SSBs and ultra processed foods, will continue to increase and physical activity will continue to decline [31], (full details in S1 File).

These projections allow us to construct a baseline estimate that is realistic and in line with trends in other countries with similar risk factor profiles showing an increase and then stabilising in diabetes incidence [54]. Diabetes prevalence in adults is projected to increase from around 12.5% of adults in 2020 to 15.4% in 2030 and 20.9% by 2050. Obesity prevalence in adults is projected to increase from 29.2% in 2020 to 39.2% by 2050. Women have three times the prevalence of obesity in 2020 than men (38.4% versus 12.5%, respectively), but that

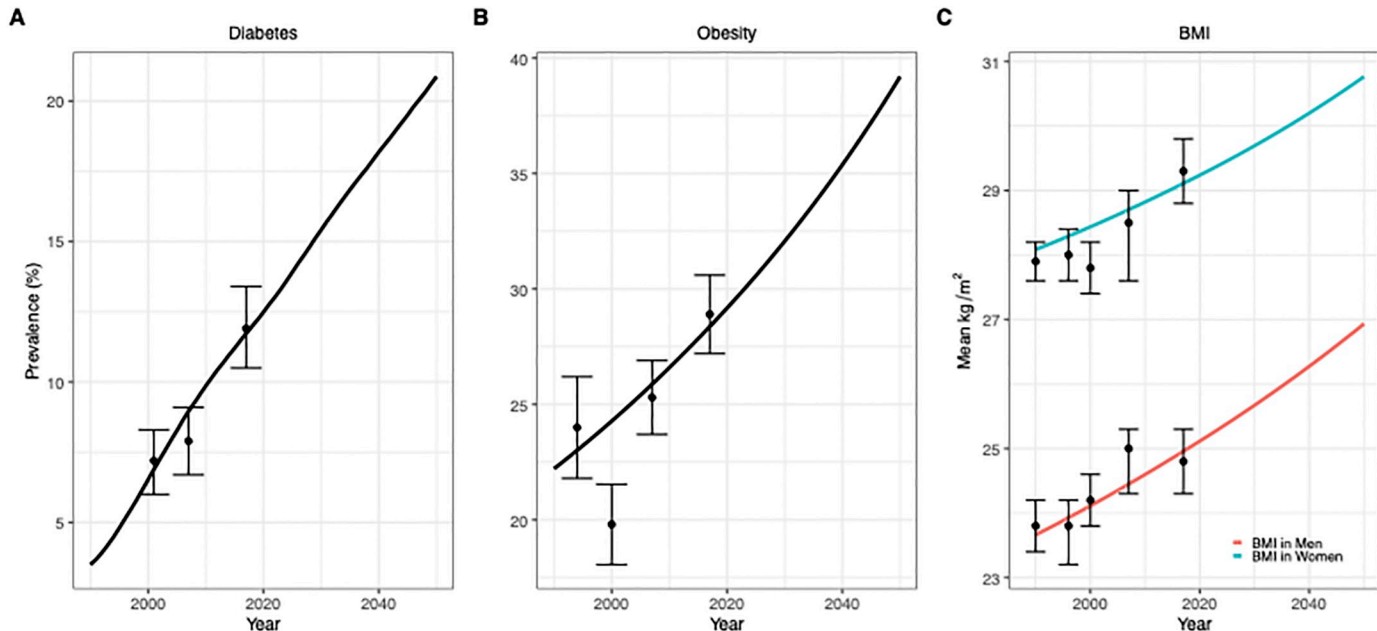

**Fig 2. Comparison of modelled outcomes (diabetes prevalence (A), obesity prevalence (B), and body mass index (C)) and projections to 2050 to survey measured estimates in adults (20+ years with 95%CI) from Jamaica(19–22).**

difference is expected to reduce by 2050 to 48.8% obesity prevalence in women to 22.2% obesity prevalence in men.

## Scenario testing: Achieving global targets

We modelled possible magnitude changes in dietary intake and in physical activity necessary to achieve the WHO Global Action Plan [3] targets set for 2025 (i.e., stop the increase in prevalence of diabetes and obesity among adults using 2010 as the baseline). We assumed a start date for change from 2010. The results of these scenarios are presented in Fig 3a and 3b.

All the scenarios assume trends for ageing and caloric intake will continue increasing, and physical activity will continue decreasing so only a magnitude reduction in caloric intake and physical activity are modelled. The greatest change for both diabetes prevalence and obesity prevalence comes from a combined dramatic reduction in caloric intake (30% lower overall compared to baseline trends) and a tripling of the estimated total MVPA in 2010. Changes in MVPA are the most powerful determinants decreasing the prevalence of diabetes both via obesity reduction and through reductions in incidence independent of weight change. Changes in caloric intake, in this case, only present a marginal decrease in diabetes prevalence (Fig 3a). If the caloric reduction was assumed to be a result of reducing SSB intake alone, then the reduction in diabetes prevalence would be more pronounced because for this model, SSB intake is the only dietary component associated with diabetes incidence independent of weight change.

For obesity prevalence, any caloric intake reductions and physical activity increases together bring substantial reductions in obesity prevalence (Fig 3b). The only scenario that could achieve the global targets proposed of stopping the rise for both obesity and diabetes prevalence over the course of 15 years is the most intensive scenario including a tripling of MVPA and an overall 30% reduction in caloric intake for the population in 2010 (assuming the relative trend in decreasing physical activity and increasing caloric intake are maintained).

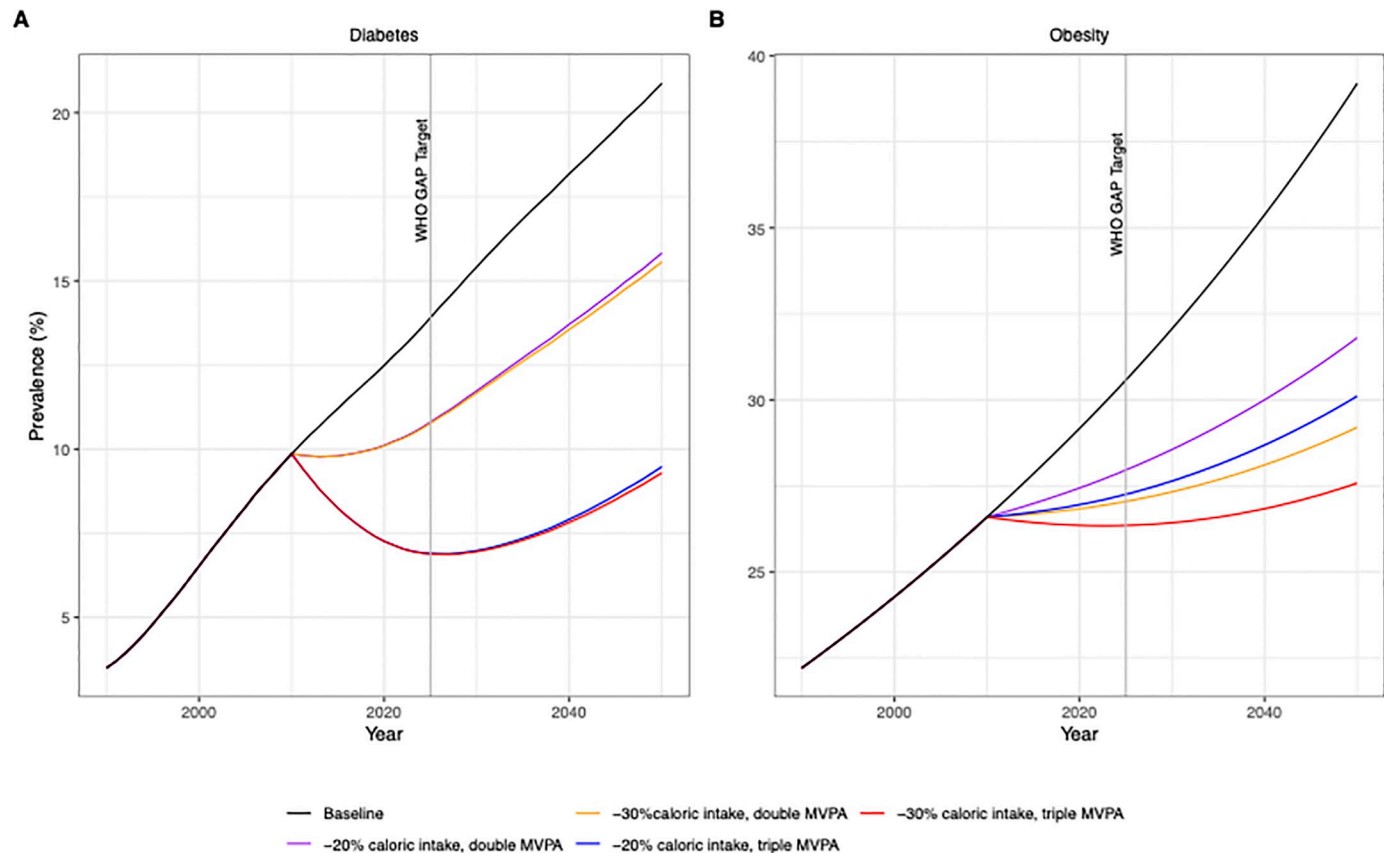

**Fig 3. Scenarios for the magnitude of change in physical activity and caloric intake necessary to achieve global targets for diabetes (A) and obesity (B) prevalence (in adults 20+ years).**

Alternative trends for caloric intake and physical activity are discussed in detail in the S1 File. Briefly, when we assume no change in the trends of diabetes and obesity over time, global targets could still only be achieved by tripling the estimated MVPA in 2010 and a 30% reduction in caloric intake levels from 2010. For diabetes prevalence, a tripling of MVPA and a 20% reduction in caloric intake also achieved a magnitude of change to meet the target of no increase in prevalence from 2010 to 2025. In analysis, tripling MVPA above the baseline predicted trend alone was enough to achieve a reduction in diabetes prevalence without changes in caloric intake, but only changes in both determinants could reduce obesity prevalence.

## Scenario testing: Scenarios relevant to the Caribbean

The scenario combinations and individual interventions tested are presented in Table 2, with corresponding future impacts on diabetes and obesity prevalence in adults presented in Fig 4a and 4b and Table 3.

## Impact of combined interventions on diabetes prevalence

An intensive campaign that includes a downstream health system component targeting those living with pre-diabetes is most effective at reducing diabetes prevalence. The change in diabetes prevalence from this scenario results in an absolute decrease of 2 percentage points (p.p) in

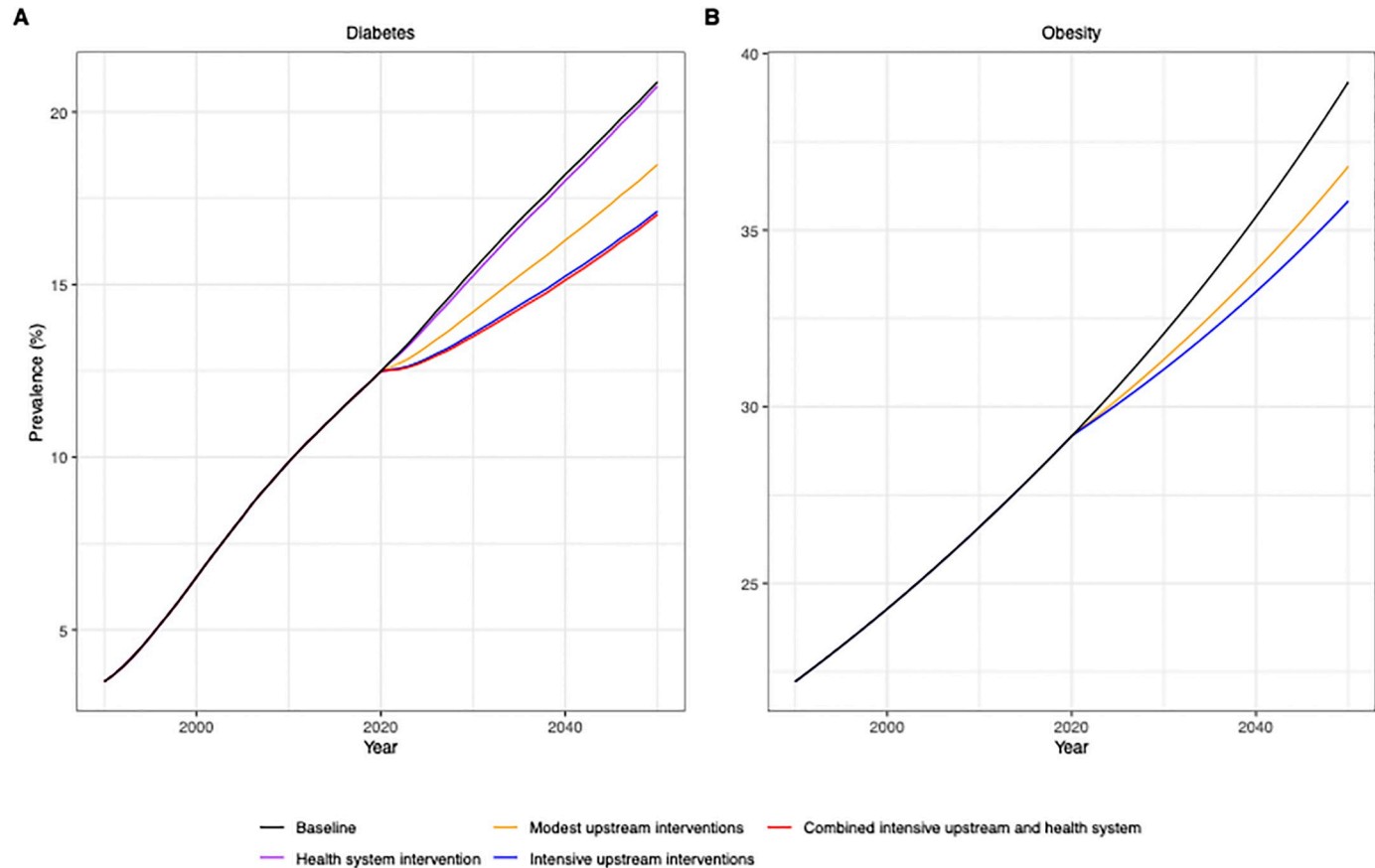

**Fig 4. Effects of scenarios for intervention relevant to the Caribbean on diabetes (A) and obesity (B) prevalence in adults (20+ years).**

10 years and 3.9 p.p. over 30 years. But even the modest intensity scenario yields a decrease of 1.2 p.p. in 10 years and 2.3 p.p. in 30 years. The upstream prevention scenarios, including providing diabetes self-management education or intensive lifestyle modification to those at high risk (see Table 2), would also reduce all-cause mortality in people with diabetes beyond what an exclusively downstream intervention can do. Mortality rates in people with diabetes by 2050 would be reduced by 12% in the moderate intensity upstream scenario compared to 8% with the downstream scenario. A combined approach with intensive upstream and a downstream intervention could prevent deaths in people with diabetes by 2050 by 25%.

## Impact of combined interventions on obesity prevalence

The scenarios tested have a modest impact on obesity prevalence (Fig 4b). The greatest reduction in obesity prevalence is from the intensive upstream intervention. This scenario results in an absolute decrease in obesity prevalence of 1 p.p. by 2030 and 3.4 p.p. by 2050 compared to the baseline projections. However, the gains from an intensive upstream intervention are only marginally better than one from a moderate upstream intervention which also reduces obesity prevalence by 0.73 p.p. by 2030 and 2.4 p.p. in 2050. Downstream interventions were not included in the modelling for obesity prevalence because the population affected was too small to be reflected in the model.

**Table 3. Change in diabetes and obesity prevalence from combined scenarios compared to the baseline prediction.**

| Scenario | Change in diabetes prevalence—p.p. (%) | | | |
|---|---|---|---|---|
| | 2021 | 2025 | 2030 | 2050 |
| Baseline predicted prevalence | 12.7 | 13.9 | 15.4 | 20.9 |
| Downstream | -0.06 (-0.5%) | -0.1 (-0.7%) | -0.2 (-0.7%) | -0.2 (-0.9%) |
| Modest upstream | -0.1 (-0.8%) | -0.7 (-5.0%) | -1.2 (-7.8%) | -2.4 (-11.5%) |
| Intensive upstream | -0.2 (-1.6%) | -1.0 (-7.2%) | -1.8 (-11.7%) | -3.8 (-18.6%) |
| Combined intensive upstream and downstream | -0.2 (-1.6%) | -1.1 (-7.9%) | -1.9 (-12.3%) | -3.9 (-18.7%) |
| | Change in obesity prevalence—p.p. (%) | | | |
| | 2021 | 2025 | 2030 | 2050 |
| Baseline predicted prevalence | 29.4 | 30.6 | 32.1 | 39.2 |
| Modest upstream | -0.03 (-0.1%) | -0.4 (-1.3%) | -0.8 (-2.5%) | -2.4 (-6.1%) |
| Intensive upstream | -0.06 (-0.2%) | -0.5 (-1.6%) | -1.0 (-3.1%) | -3.4 (-8.7%) |

## Impact of individual interventions on diabetes prevalence

Besides testing scenarios for combined interventions, we also assessed the impact of the individual interventions from Table 2 on diabetes and obesity prevalence (see Table 4). The highest impact from any single intervention to prevent diabetes comes from reductions in sugar

**Table 4. Impact of individual changes in dietary patterns and physical activity on diabetes and obesity prevalence.**

| | Scenario | Change in diabetes prevalence p.p. (%) | | | |
|---|---|---|---|---|---|
| | | 2021 | 2025 | 2030 | 2050 |
| | Baseline predicted prevalence | 12.7 | 13.9 | 15.4 | 20.9 |
| Downstream | Lifestyle modification intervention to 25% of pre-diabetes population | -0.05 (-0.4%) | -0.22 (-1.5%) | -0.37 (-2.3%) | -0.69 (-3.3%) |
| | Diabetes self-management education to 30% of people with diabetes | +0.02 (+0.17%) | +0.1 (+0.7%) | +0.20 (1.3%) | +0.57 (2.7%) |
| Modest intensity | Additional 15 minutes/day of MVPA | -0.05 (-0.4%) | -0.22 (-1.5%) | -0.37 (-2.4%) | -0.69 (-3.3%) |
| | Reduce SSB consumption by 15% | -0.06 (-0.4%) | -0.25 (-1.7%) | -0.42 (-2.7%) | -0.83 (-4.0%) |
| | Reduce ultra processed food consumption by 15% | 0 (0) | -0.002 (-0.01%) | -0.007 (-0.04%) | -0.06 (-0.3%) |
| | Increase fruit and vegetable intake by 15% | -0.002 (-0.02%) | -0.009 (-0.07%) | -0.01 (-0.1%) | -0.03 (-0.1%) |
| High intensity | Additional 30 minutes/day of MVPA | -0.10 (-0.8%) | -0.44 (-3.1%) | -0.75 (-4.8%) | -1.42 (-6.8%) |
| | Reduce SSB consumption by 25% | -0.29 (-2.2%) | -1.3 (-9.1%) | -2.3 (-14.7%) | -4.8 (-23.1%) |
| | Reduce ultra processed food consumption by 25% | 0 (0) | -0.003 (-0.02%) | -0.01 (-0.07%) | -0.10 (-0.5%) |
| | Increase fruit and vegetable intake by 25% | -0.003 (-0.3%) | -0.015 (-0.1%) | -0.02 (-0.1%) | -0.05 (-0.2%) |
| Information* | Public health mass media campaigns and front-of-package labelling | 0(0) | -0.25 (-1.79%) | -0.42 (-2.3%) | -0.82 (-3.9%( |
| | Scenario | Change in obesity prevalence p.p. (%) | | | |
| | | 2021 | 2025 | 2030 | 2050 |
| | Baseline predicted prevalence | 29.4 | 30.6 | 32.1 | 39.2 |
| Modest intensity | Additional 15 minutes/day of MVPA | -0.008 (-0.03%) | -0.042 (-0.14%) | -0.08 (-0.3%) | -0.27 (-0.7%) |
| | Reduce SSB consumption by 15% | -0.03 (-0.01%) | -0.05 (-0.02%) | -0.11 (-0.4%) | -0.36 (-0.9%) |
| | Reduce UPF consumption by 15% | -0.02 (0.06%) | -0.09 (-0.3%) | -0.18 (-0.6%) | -0.72 (-1.8%) |
| High intensity | Additional 30 minutes/day of MVPA | -0.02 (-0.08%) | -0.08 (-0.3%) | -0.17 (-0.5%) | -0.55 (-1.4%) |
| | Reduce SSB consumption by 25% | -0.05 (-0.18%) | -0.3 (-0.9%) | -0.6 (-1.7%) | -1.8 (-4.6%) |
| | Reduce UPF consumption by 25% | -0.03 (-0.09%) | -0.14 (-0.5%) | -0.3 (-1.0%) | -1.2 (-3.1%) |
| Information* | Public health mass media campaigns and front-of-package labelling | 0 (0) | -0.2 (-0.65%) | -0.4 (-1.2%) | -1.0 (-2.6%) |

* Information interventions include a combination of a mass media campaign to increase fruit and vegetable intake, nutrient warning front-of-package labelling, and a campaign to increase physical activity. We assume the effects of these campaigns are sustained over time.

sweetened beverage intake. If the intake of SSBs were decreased by at least 15% in 2020, diabetes prevalence would decrease by 2.4% or an absolute reduction of 0.37 p.p. by 2030 and 0.83 p.p. by 2050 compared to the baseline predicted prevalence. The dose-response curves (see S1 File) for reductions in SSB consumption are not linear. The greatest impact is seen in the first five years after a reduction, and that the effect for this impact tapers over time, in part due to other effects like increases in ageing. In addition, each 10% reduction in intake yields a greater, non-linear reduction in diabetes prevalence so that the benefits of a 50% reduction in intake are more than 5 times the benefits of a 10% in intake.

Of a similar magnitude, adding a mean 15 minutes of MVPA per day above the baseline for the population would lead to a 2.4% decrease in diabetes prevalence by 2030 (or 0.37 p.p.) and a 3.3% decrease by 2050. Other interventions did little to shift diabetes prevalence in the short or long term. More intensive changes yielded greater results. Predictably, adding daily MVPA to 30 min above the baseline reduced diabetes prevalence by -0.75 p.p. in 2030 and -1.42 p.p. by 2050. Reducing SSB consumption by 25% compared to the baseline also produced similar gains (-0.71 p.p by 2030 and -1.42 p.p. by 2050). Other interventions only had a modest impact on reducing diabetes.

## Impact of individual interventions on obesity prevalence

Obesity prevalence was most reduced compared to baseline predictions by reductions in caloric intake. Reducing ultra-processed food consumption by 15% led to a reduction in obesity prevalence of 0.18 p.p. by 2030 and 0.72 p.p. by 2050. If this reduction were increased to 25% compared to baseline, the reduction in obesity would be 0.3 p.p. by 2030 and 1.2 p.p. by 2050. Dose-response curves indicate a non-linear effect on obesity so that every 10% reduction in intake produces greater decreases in prevalence, but unlike with diabetes prevalence, the magnitude of those decreases is greater over time, especially for the highest proportional decrease in intake. In other words, the gains from a 50% decrease in ultra-processed foods after 30 years are more than five times greater than the gains from a 10% decrease over time (see S1 File).

Adding minutes of MVPA above the current baseline was also effective in reducing obesity prevalence. Adding 15 minutes/day of MVPA decreased obesity prevalence by 0.08 p.p. in 2030 and 0.27 p.p. in 2050. The dose-response relationship of MVPA to obesity prevalence was found to be almost linear so that doubling MVPA to an additional 30 minutes/day above the baseline reduced prevalence by 0.17 p.p. in 2030 and 0.55 p.p. in 2050.

In all, individual scenarios all resulted in much smaller gains than those of the combined scenarios.

## Discussion

The alarming rise of type 2 diabetes and obesity in the Caribbean has placed it firmly on the political agenda since at least the Port-of-Spain Declaration in 2007 [13]; however, the prevalence of diabetes and obesity continue to rise in the region [14]. The findings from this model show that for Jamaica, like many parts of the world [55], global targets proposed by the WHO Global Action Plan [3] to stop the rise in obesity and diabetes prevalence by 2025 are unrealistic. The magnitude of change necessary to achieve these targets in the modifiable determinants of obesity (in particular diet and physical activity) are unachievable, especially over the timeframe proposed by the Global Plan. No country in the world has managed to reduce obesity or diabetes prevalence, barring those affected by natural disasters or humanitarian crises [56–58].

### Rethinking global targets

Global targets based on reducing or stabilising prevalence may be unattainable. In a disease where the duration is lifelong, like many NCDs, it is possible to reduce the incidence of a disease yet continue to see a rise in prevalence. Without an outflow that is larger than the inflow, prevalence will continue to rise. For NCDs like diabetes, there is no effective outflow at a population level besides mortality. Studies suggest that with an intensive lifestyle intervention aimed at dietary restriction and weight loss it may be possible to reverse diabetes in individuals [6, 59, 60], but this is not feasible on a large scale in a developing country at the moment. In addition, any interventions aimed at reducing mortality, such as providing diabetes self-management education will prevent deaths and reduce the mortality outflow. This is obviously a good outcome, but will increase diabetes prevalence and thus work against achieving a target based solely on reducing prevalence. Instead, targets should focus on several indicators that track the benefits of improvements over a range of outcomes that include incidence, quality of life, mortality, and rates of complications., Such an approach is supported by Gregg et al. who recommended monitoring targets for the Global Diabetes Compact [61], acknowledging that reliably assessing trends in these is often more challenging and resource intensive than measuring prevalence. The most recent report of the WHO Global Monitoring Framework from 2019 shows progress on implementing NCD guidelines and monitoring for outcomes, but only 53% of countries reported having the six essential technologies for monitoring (including anthropometric measurement, blood glucose, blood pressure, and cholesterol) available routinely in primary care [62].

### Interventions to reduce the burden

Strategies that focus on the prevention of obesity and diabetes show the greatest long-term success in both diabetes prevalence and all-cause mortality for people with diabetes. When we examine the more plausible scenarios, we find that even a more modest change in dietary intake and physical activity has a substantial long term impact on diabetes prevalence compared to the baseline predicted level and also reduces obesity, although to a lesser extent. These changes are not easily achieved, but some interventions have shown promise.

For physical activity, improving infrastructure for active transport and leisure has been shown to increase MVPA in sedentary populations [63–66] as well as developing reliable public transportation systems [67]. Trends in physical activity suggest the greatest losses in MVPA are occurring in occupational physical activity in developing countries [31], and those losses are unlikely to be recovered as jobs become more sedentary. As a result, interventions often focus on increasing leisure-time physical activity where even a small change for sedentary individuals can lead to a reduction in risk for diabetes independent of weight loss [43]. In addition, replacing sedentary time with light physical activity which can include short walks can also lead to weight loss and has been linked to reductions in diabetes incidence [68, 69]. The benefits of increasing physical activity are much wider and include decreasing all-cause mortality, cancer risk reduction, and mental health benefits [70].

There are a number of possible interventions for achieving reductions in caloric intake. Information campaigns including mass media public health education campaigns and front-of-package food labelling have been shown to be effective in calorie reduction [71, 72], Similarly, reducing SSB intake, for which the Caribbean has the highest rate [37] can have a substantial effect on both obesity and diabetes prevalence over time. The Caribbean has made taxing SSBs a priority, but the structure and implementation of those taxes must be considered to achieve a price difference that will effectively reduce intake [73–75]. Taxes on unhealthy foods similar to those implemented in Chile [76] or in combination with public health

campaigns as was done in Mexico [73] are used in the model and might achieve a 20% decrease in intake for the population. In our model, SSB intake was one of two dietary components (with fruit and vegetable intake) associated with diabetes incidence independent of weight gain. Thus, if the scenarios presented in Fig 3a focused on achieving the same calorie reduction purely through a reduction in SSB intake, the impact on diabetes prevalence would likely be greater than what is shown here.

Fruit and vegetable intake in the Caribbean is very low [77]. Increasing that intake could also improve outcomes and likely with a greater magnitude than what the model shows. There is some evidence that overall energy intake does not decrease when including more fruits and vegetables in the diet, but that weight loss can still occur [78, 79]. The mechanisms by which fresh fruits and vegetable intake affects metabolism may also reduce diabetes incidence [80]. Similarly, reducing ultra processed foods may give benefits beyond weight reduction, and also reduce the incidence of diabetes [81, 82]. Our findings suggest that from a policy perspective, none of these interventions should be attempted in isolation. Rather, a suite of coordinated policies like those implemented in Chile, show the greatest success in shifting dietary patterns across various domains [76]. Downstream interventions in the health system for people at high risk or with diabetes are an important component that can greatly improve the quality of life and reduce disability in those affected [83, 84]. However, it is clear from these scenarios that intervening exclusively downstream is not an effective prevention strategy at the population level.

The commercial determinants of health refer to a broad range of private sector activities that affect the health of populations [85]. Some of the interventions presented here respond to aspects of the commercial determinants of health including price changes on ultra processed foods and front-of-package labelling [86], but other key areas, such as the potential for more comprehensively addressing marketing by producers of unhealthy food were not addressed. This is in part due to a lack of clear evidence on the effects of regulation, particularly for adults. There is good evidence that regulation of marketing to children has the potential to reduce unhealthy diets [87]. Nevertheless, any comprehensive policy action on obesity should include limitations on marketing and make efforts to reduce the reach and influence of the ultra processed food industry [88].

## Targeted interventions

Health systems represent for many countries the first sector to engage in prevention of diabetes. In controlled trials for people at high risk for type 2 diabetes, intensive lifestyle interventions have been shown to be highly effective with lasting benefits [5]. However, when taking into account the small number of people that are engaged in these trials and the challenges of adherence to protocols, our analysis shows these types of interventions give only marginal benefits to the population as a whole. It is important to note that those interventions can provide a great benefit to the people that are engaged. A coordinated strategy for preventing new cases of obesity and diabetes together with a health system strategy to mitigate the effects of those diseases is the most sound approach to achieve many desirable health outcomes beyond just a reduction in prevalence.

For the Caribbean, most populations, including in Jamaica, have a strong gender disparity in physical inactivity and obesity where women have almost three times the rate compared to men [25]. The reasons for the disparities are complex [89] and any intervention targeted to women must take into account this complexity. In addition, a systematic review found that diabetes was commoner among those with a lower education that those with higher education

in the Caribbean [25], a social determinant of health, suggesting targeting of interventions to the most vulnerable can be tailored in different ways.

This model shows that intervening in physical activity or obesity reduction for women would have a clear impact on overall population measures of diabetes and obesity and may be an effective targeted approach, especially where resources may be limited. Stakeholders have also focused on targeting interventions to children and youth in the hopes of preventing new cases of obesity and NCDs in adulthood. Any opportunity for prevention can help reduce the burden of NCDs, but it is important to consider that an intervention targeted at youth will take decades to show results [90–92]. Moving away from an obesogenic environment for all of society can ensure those gains made early in life are not lost in adulthood.

## Limitations

Like any model, this system dynamics model is a simplification of reality. It is intended as a tool for understanding the possible magnitude of changes in the outcomes of interest (diabetes and obesity prevalence) by intervening in different points. It is not intended, however, to provide a precise prediction of diabetes prevalence in the near or long term, although we did calibrate the model carefully to reflect the historical data available for Jamaica. We assumed changes from interventions would be apparent within a one year time step, which may be an oversimplification. Interventions can take longer to show results.

One of the most important assumptions underlying the projections in prevalence for obesity and diabetes are the underlying trends in physical activity, caloric intake, and ageing of the population. We rely on published projections of similar plausible models for population ageing [26] and physical activity [31]. Trends for caloric intake were a combination of projections for SSB intake [37], ultra processed food consumption projections [12, 38], and assuming trajectories continue in the next decades. We felt these trends accurately reflected stakeholder input that described reinforcing feedback loops shifting social norms towards more physical inactivity and increasing caloric intake [17, 18]. However, the data on trends in these determinants are lacking for the region and the true trajectories may be different. The literature used to underlie our assumptions do not support a halt to the rise in obesity and diabetes prevalence in the future, nor is there evidence for a decrease in caloric intake or an increase in physical activity at the population level. It is possible that there may be some threshold for which physical activity and caloric intake stabilise, although it is unclear where that level may be. However, we acknowledge that other system dynamics models exploring the obesity epidemic in the United States have noted that in terms of caloric balance, assuming a linear future trend may exaggerate future prevalence estimates [49, 93], and this may be the case in our modelling.

There are a number of places where other simplifications may be underestimating the true impact of changes. For instance, dietary intake patterns can have multiple and synergistic effects on weight reduction and obesity. There is some evidence that shows that consumption of ultra processed foods can lead to excessive caloric intake [81] which means that the effect of reducing these may be underestimated in the model. We did not take into account a detailed dietary pattern including major macronutrients like fat content, salt, or consumption of nuts, all which may have an association with changing weight and diabetes incidence [42]. We also did not take into account the distribution of determinants in the population which in many cases are not normally distributed. For instance, from the self-reported data we do have [19, 20], it is clear that a small portion of highly active individuals may be skewing the mean level of physical activity of a largely sedentary population. In situations such as these, we used a linear relationship for the reduction in diabetes incidence from increases in mean MVPA (see S1

File), but it is possible that intervening with the least active would be the most effective way to shift population-level physical activity.

The model includes the major risk factors for type 2 diabetes (obesity, aspects of diet, and physical inactivity) but there are many other factors that may influence the risk and onset of type 2 diabetes that were not included. For instance, we did not include rates of hypertension, tobacco use (which is relatively low in the English-speaking Caribbean and highly gendered, being commoner in men) [94], or alcohol use which are also known to directly impact diabetes incidence to a lesser extent and for which cost-effective solutions can be implemented [84]. A recently published meta-analysis suggests that some types of antihypertensives can be effective in preventing type 2 diabetes [95]. Finally, it is also possible that the relative impact of obesity on the onset of diabetes is greater for this population than what we assume in the model [96]. If that is the case, any change in obesity is likely to have an even greater effect on reducing diabetes incidence and prevalence.

### Systems thinking for policy and beyond

Our analysis of the global targets proposed in the WHO Global Action Plan for 2025 show them to be overly ambitious if not unachievable. Any future targets to monitor progress on how policies reduce the burden of diabetes and obesity should take into account the complex dynamics including accumulations, feedbacks, and a long time horizon for achieving change in prevalence. Diabetes can also be conceptualised not as a binary condition but as a continuum of risk for complications where it may be appropriate to intervene in different ways at different stages. Any gains shown from interventions presented may be an underestimate as there may be synergistic effects not accounted for in the model. Furthermore, the benefits of reducing obesity go far beyond reductions in diabetes prevalence and can influence quality of life, mental health, cancer risk, cardiovascular disease, and a number of other conditions [97]. Even small relative reductions compared to baseline trends are therefore worth the effort to obtain. This model should help to set realistic goals for policy makers and reinforce the importance of mobilising and coordinating policy across several sectors and using many different approaches at once. Financing and supporting those goals remain a challenge for resource-limited countries, like Jamaica. The CARICOM placed the public health NCD crisis in the region as a key component of a Ten Point Plan for Reparatory Justice for formerly enslaved populations, arguing that there is a direct link between the current high levels NCDs and the historical brutalities of the colonial experience in the Caribbean, including Jamaica [98]. NCD prevention and control are increasingly being seen as critical components of the global health agenda [99] but continue to be underfunded [100].

The scenarios tested here are ambitious, but they echo what many of the policymakers in Jamaica and the region have already known that the whole of the system must be considered and sectors must be engaged to reduce the burden of NCDs. Finally, achieving reductions in NCDs and their determinants is not a goal for a single political cycle, but something for which countries should be prepared to engage in over the long term and for the whole of society.

### Supporting information

**S1 File.**
(PDF)

## Acknowledgments

The authors would like to gratefully acknowledge the participation of key stakeholders in the conceptual development of the causal loop diagrams underlying the assumptions of the model presented here. Their input is invaluable. We would also like to acknowledge the support of staff at the University of the West Indies George Alleyne Chronic Disease Research Centre for their top notch administrative and organisational support.

## Author Contributions

**Conceptualization:** Leonor Guariguata, Natasha Sobers, James Woodcock, T. Alafia Samuels, Cornelia Guell, Nigel Unwin.

**Data curation:** Leonor Guariguata.

**Formal analysis:** Leonor Guariguata, Leandro Garcia.

**Funding acquisition:** Nigel Unwin.

**Investigation:** Leonor Guariguata, Natasha Sobers, Trevor S. Ferguson, James Woodcock, T. Alafia Samuels, Cornelia Guell, Nigel Unwin.

**Methodology:** Leonor Guariguata, Leandro Garcia, Trevor S. Ferguson, James Woodcock, T. Alafia Samuels, Cornelia Guell, Nigel Unwin.

**Project administration:** Leonor Guariguata.

**Resources:** Nigel Unwin.

**Software:** Leonor Guariguata.

**Supervision:** T. Alafia Samuels, Cornelia Guell, Nigel Unwin.

**Validation:** Leandro Garcia, Natasha Sobers, Trevor S. Ferguson, James Woodcock.

**Visualization:** Leonor Guariguata, Leandro Garcia, James Woodcock.

**Writing – original draft:** Leonor Guariguata.

**Writing – review & editing:** Leonor Guariguata, Leandro Garcia, Natasha Sobers, Trevor S. Ferguson, James Woodcock, T. Alafia Samuels, Cornelia Guell, Nigel Unwin.

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
