## [Decision Letter · Decision Letter 0]

7 Mar 2022

PGPH-D-22-00092

Exploring ways to respond to rising obesity and diabetes in the Caribbean using a system dynamics model

Dear Dr. Unwin,

Thank you for submitting your manuscript to PLOS Global Public Health. After careful consideration, we feel that it has merit but does not fully meet PLOS Global Public Health’s publication criteria as it currently stands. Therefore, we invite you to submit a revised version of the manuscript that addresses the points raised during the review process.

We look forward to receiving your revised manuscript.

Kind regards,

Peter Rohloff

Academic Editor

Journal Requirements:

1. Please provide separate figure files in .tif or .eps format only.  Please ensure that all files are under our size limit of 20MB.  

For more information about how to convert your figure files please see our guidelines: Once you've converted your files to .tif or .eps, please also make sure that your figures meet our format requirements

2. We have noticed that you have uploaded supporting information but you have not included a list of legends.  Please add a full list of legends for all supporting information files (including figures, table and data files) after the references list. 

Additional Editor Comments (if provided):

Reviewers' comments:

Reviewer's Responses to Questions

**Comments to the Author**

1. Does this manuscript meet PLOS Global Public Health’s publication criteria? Is the manuscript technically sound, and do the data support the conclusions? The manuscript must describe methodologically and ethically rigorous research with conclusions that are appropriately drawn based on the data presented.

Reviewer #1: Yes

Reviewer #2: Yes

2. Has the statistical analysis been performed appropriately and rigorously?

Reviewer #1: Yes

Reviewer #2: Yes

3. Have the authors made all data underlying the findings in their manuscript fully available (please refer to the Data Availability Statement at the start of the manuscript PDF file)?

Reviewer #1: Yes

Reviewer #2: Yes

4. Is the manuscript presented in an intelligible fashion and written in standard English?

Reviewer #1: Yes

Reviewer #2: Yes

5. Review Comments to the Author

Reviewer #1: This is an impressively comprehensive modeling exercise to assess the feasibility of achieving global targets for diabetes prevalence and mortality, and more broadly to assess likely trends and intervention impacts in Jamaica over the next 30 years. I recommend acceptance. One element that you may want to consider in the introductory remarks or discussion is the social determinants of health (particularly poverty), as well as the specificity of Jamaica's history, and the impacts these may have on current diabetes patterns. Diabetes has a strong relationship to poverty, and both diabetes and poverty track with formerly colonized and enslaved status in the Americas. While Britain paid slave owners for the inconvenience of abolition in 1834, it has not paid reparations to the Jamaican descendants of enslaved people - worth 7.6 billion pounds today if they were to get the same deal as the slave owners. Given the enormous investments in upstream and downstream programs necessary to level off diabetes prevalence, this British responsibility could prove very relevant.

Overall, this is excellent, thorough modeling and the paper is well-written.

Reviewer #2: This manuscript presents the results of a modeling exercise in Jamaica to assess trends and policy strategies to address obesity and diabetes in Jamaica. This study builds upon a previously published systems dynamic model with a high level of adaptation to Jamaica. The authors' top-line finding is that targets put forth in the WHO Global Action Plan are totally unrealistic. They also suggest specific measures both upstream and downstream which are likely to make the biggest difference in the future levels of obesity and diabetes in Jamaica. Overall, I found this study to be methodologically rigorous and well-conducted, and I think it should be published in PLOS Global Public Health.

Major comments:

- What I liked most about this study was how it used a well-known modeling platform and then did a lot of formative work to tailor the models to Jamaica and also to incorporate the best-possible input data into the model. It is very impressive.

- I am wondering if the authors might include a brief summary of how the model works beyond the model figures. I am not a quantitative modeler and am unfamiliar with systems modeling. But I am a reasonably average consumer of quantitative modeling papers. I found it hard to understand how the simulation was being performed, whether it was a Markov (assume not), whether uncertainty was incorporated. Are these based on regressions? Pardon my ignorance but a bit more statistical detail for a non-modeler would be useful in the "model implementation" section, rather than referring to the software used. Forgive me if I missed it, but I did not these details in the appendix.

- I am confused if the model reports diabetes-related deaths and whether this is part of the present manuscript. I note that death is included in the model figure and is part of the original AJPH paper using this model. For example, the paragraph at the top of page 17 has quite a bit of language in the results section describing the model's impact on death, but it is not clear to me if this is referring to the model results or making an interpretation based on the model assumptions in Table 2. I would ask the authors to either present the death estimates, if they have them, or to clarify if estimated deaths will be reported in a separate publication.

Minor comments:

- Might the authors consider reducing the number of abbreviations to enhance clarity and readability?

- I recommend resizing the axes of Figure 2; the vertical orientation makes it a bit difficult to interpret.

- Thank you for making the code publicly available.

- Some text in results may be better suited for the methods section either in the full text or appendix. (For example line 311: "Scenarios for obesity reduction do not include any downstream interventions like bariatric surgery because there is no evidence that these have a measurable impact at the population level.")

- I found the structure of the results section quite difficult to follow. Do the italics refer to both headings and also subheadings? It would be helpful to more clearly differentiate which text refers to which heading level and to keep more consistent labeling.

- Line 406 referencing diabetes reversal, I would suggest referencing the results of the DiRECT study to strengthen this paragraph

Lean MEJ, Leslie WS, Barnes AC, et al. Primary care-led weight management for remission of type 2 diabetes (DiRECT): an open-label, cluster-randomised trial. The Lancet 2018; 391(10120): 541-51.

Lean MEJ, Leslie WS, Barnes AC, et al. Durability of a primary care-led weight-management intervention for remission of type 2 diabetes: 2-year results of the DiRECT open-label, cluster-randomised trial. The Lancet Diabetes & Endocrinology 2019; 7(5): 344-55.

- Lines 414-418, these are very nice and well-thought-out points about monitoring and targets at the national level. Are there any examples of countries that have done this successfully that you could reference? Might also be useful to consider how the new draft WHO report on global diabetes targets fits into the paradigm you propose:

https://www.who.int/publications/m/item/improving-health-outcomes-of-people-with-diabetes-mellitus

- This paper does an excellent job throughout of citing appropriate literature and positioning its results within the broader landscape of prior studies. At the same time, I wonder if there is a role for simplifying or abbreviating some of the elements to enhance readability. For example, in my estimation, the discussion is approximately 3,000 words, and the limitations section is approximately 1,000 words itself.

6. PLOS authors have the option to publish the peer review history of their article (what does this mean?). If published, this will include your full peer review and any attached files.

**Do you want your identity to be public for this peer review?** For information about this choice, including consent withdrawal, please see our Privacy Policy.

Reviewer #1: No

Reviewer #2: No

---

## [Editor Report · Decision Letter 1]

7 Apr 2022

Exploring ways to respond to rising obesity and diabetes in the Caribbean using a system dynamics model

PGPH-D-22-00092R1

Dear Professor Unwin,

We are pleased to inform you that your manuscript 'Exploring ways to respond to rising obesity and diabetes in the Caribbean using a system dynamics model' has been provisionally accepted for publication in PLOS Global Public Health.

Best regards,

Peter Rohloff

Academic Editor